# Spin current driven by ultrafast magnetization of FeRh

Kyuhwe Kang[1], Hiroki Omura[2], Daniel Yesudas[1], OukJae Lee [3], Kyung-Jin Lee [4], Hyun-Woo Lee [5], Tomoyasu Taniyama[2] & Gyung-Min Choi [1,6] ✉

Laser-induced ultrafast demagnetization is an important phenomenon that probes arguably the ultimate limits of the angular momentum dynamics in solid. Unfortunately, many aspects of the dynamics remain unclear except that the demagnetization transfers the angular momentum eventually to the lattice. In particular, the role and origin of electron-carried spin currents in the demagnetization process are debated. Here we experimentally probe the spin current in the opposite phenomenon, i.e., laser-induced ultrafast magnetization of FeRh, where the laser pump pulse initiates the angular momentum build-up rather than its dissipation. Using the time-resolved magneto-optical Kerr effect, we directly measure the ultrafast-magnetization-driven spin current in a FeRh/Cu heterostructure. A strong correlation between the spin current and the magnetization dynamics of FeRh is found even though the spin filter effect is negligible in this opposite process. This result implies that the angular momentum build-up is achieved by an angular momentum transfer from the electron bath (supplier) to the magnon bath (receiver) and followed by the spatial transport of angular momentum (spin current) and dissipation of angular momentum to the phonon bath (spin relaxation).

Ultrafast demagnetization is a rapid quenching of the magnetic ordering in a ferromagnetic metal (FM) in less than a picoseconds[1–4]. Such a short timescale indicates a rapid dissipation mechanism for angular momentum in FM. Considering the Einstein-de Haas effect, the ultimate destination of angular momentum should be the lattice bath[5–7]. A recent study in a single FM layer demonstrated that ultrafast demagnetization induces a circularly polarized phonon in less than a picosecond, suggesting a fast transfer of angular momentum between the magnetization bath and phonon bath[8]. However, the microscopic process of the angular momentum transfer remains poorly understood. In particular, the role of the electron-carried spin current in the transfer process remains unclear.

The electronic contribution to the angular momentum transfer was revealed from the spin dynamics in heterostructures: ultrafast demagnetization of FM generates a transient spin current in a nonmagnetic metal (NM). Ultrafast-demagnetization-driven spin currents have been confirmed by various experimental observations, such as spin accumulation on NM in FM/NM[9–13], terahertz generation from NM in FM/NM[14–17], coupling of demagnetization dynamics of FM layers in FM/NM/FM with the collinear magnetization[18–20], and spin-transfer-torque on FM in FM/NM/FM with a non-collinear magnetization[21,22]. However, the mechanism of the spin current remains controversial[9,10,23–25]. The superdiffusive theory is hot-electron version of the spin filter effect and argues that the spin-dependent transport of hot electrons inside FM generates a strong spin current to NM[23,24]. Since the electronic density-of-states of FM is spin-dependent, an electronic transport at the FM/NM interface leads to a spin filter effect[26]. Another mechanism is the angular momentum transfer between magnons (wave-like excitation of local magnetic moments) and conduction electrons[4,10,27,28]. The angular momentum of the FM

[1]Department of Energy Science, Sungkyunkwan University, Suwon 16419, Korea. [2]Department of Physics, Nagoya University, Nagoya 464-8602, Japan. [3]Center for Spintronics, Korea Institute of Science and Technology, Seoul 02792, Korea. [4]Department of Physics, Korea Advanced Institute of Science and Technology, Daejeon 34141, Korea. [5]Department of Physics, Pohang University of Science and Technology, Pohang 37673, Korea. [6]Center for Integrated Nanostructure Physics, Institute for Basic Science, Suwon 16419, Korea. ✉e-mail: gmchoi@skku.edu

phase is stored in the $d$-band electrons, which are responsible for the magnetization, and angular momentum transfer should be mediated by the magnetic excitations of the $d$-band, such as magnons[27,28] and Stoner excitations[4] (in this work, for simplicity, we call magnons as a representative of the magnetic excitations). When the angular momentum in the magnon bath ($d$ band) is converted to the angular momentum in the electron bath ($sp$ band), spin current is generated by $-dM/dt$, where $M$ is the magnetization of the magnon bath, and $t$ is the time ($dM/dt$ model)[10].

In this study, we investigated the spin current in the reverse process, i.e., ultrafast magnetization of FeRh during the phase transition from the antiferromagnetic metal (AFM) phase to the FM phase. Whereas ultrafast demagnetization releases angular momentum to the surrounding, ultrafast magnetization of FeRh should absorb angular momentum from the surrounding. Since the spin filter effect is not allowed with the initial AFM phase, ultrafast magnetization is an optimal circumstance to investigate the mechanism for the spin current generation.

## Results and discussion

### Angular momentum transfer during phase transition

Angular momentum transfer is also important to understand the phase transition mechanism. FeRh exhibits a unique phenomenon of the 1st-order phase transition, which leads to a magnetic transition from the AFM phase to the FM phase accompanied by a lattice expansion of ~1%, at a critical temperature of ~350 K[29,30]. Previous reports have demonstrated that the magnetization change and lattice expansion can occur on the order of picoseconds during the phase transition[31–38]. However, the mechanism for such a fast phase transition is under debate. The timescale of the phase transition consists of two contributions: the timescale for the driving force responsible for the phase transition and the timescale for angular momentum transfer. As for the driving force, Kittel proposed that drastic lattice expansion drives the sign inversion of the exchange coupling parameter[39], then its timescale may be characterized by the speed of lattice expansion[35,38]. Other possible sources of the driving force include electric band structure change[37,40], magnetic moment of Rh

atom[32], and magnon excitations[41], whose timescale are expected to differ from that of lattice expansion. As for the angular momentum transfer, the driving force should initiate a rapid transfer of the angular momentum considering the huge difference in the magnetization density between the AFM and FM phases. However, the exact procedure of the angular momentum transfer during the phase transition is not known. Especially, the role of conduction electrons for the angular momentum transfer has not been studied.

In this study, we investigate angular momentum transfer during the phase transition of FeRh by measuring the ultrafast-magnetization-driven spin current in the MgO substrate/FeRh (20 nm)/Cu (120 nm) structure. The FeRh and Cu layers were grown by Molecular Beam Epitaxy (see Methods), and the FeRh/Cu interface has a clean and flat morphology (Fig. 1a). From the quasi-static measurements of the magnetization as a function of temperature, the phase of our FeRh changes from AFM to FM at ~370 K (Fig. 1b and Supplementary Sections 1 and 2). The transient dynamics of the phase transition is investigated by using an optical pump-probe technique (see Methods and Supplementary Section 3). When we inject a pump pulse on FeRh through the MgO substrate, it triggers the phase transition of FeRh. A probe pulse detects the magnetization ($\Delta M$) of FeRh and spin accumulation ($\Delta S$) of Cu via the magneto-optical Kerr effect. A probe also detects the lattice expansion ($\Delta L$) of FeRh via the strain-induced reflectivity change (Fig. 1c). We examine the correlation between the magnetization dynamics of FeRh and the spin accumulation of Cu in terms of sign, time delay, and magnitude.

### Ultrafast magnetization of FeRh

Firstly, we measure ultrafast magnetization of FeRh during the phase transition. With an initial AFM phase at a base temperature of 300 K, a pump pulse triggers an ultrafast phase transition from the AFM phase to the FM phase, and a probe pulse detects the time evolution of $\Delta M$. An external magnetic field of 0.15 T is applied to set the direction of the magnetic moment of the FM phase (see Methods). With a pump fluence >2 J m$^{-2}$, a sharp increase of $\Delta M$ occurs at 2-4 ps (Fig. 2a). However, with a pump fluence of 0.9 J m$^{-2}$, $\Delta M$ becomes negligible. Such a threshold is a characteristic behavior for the 1st order phase transition

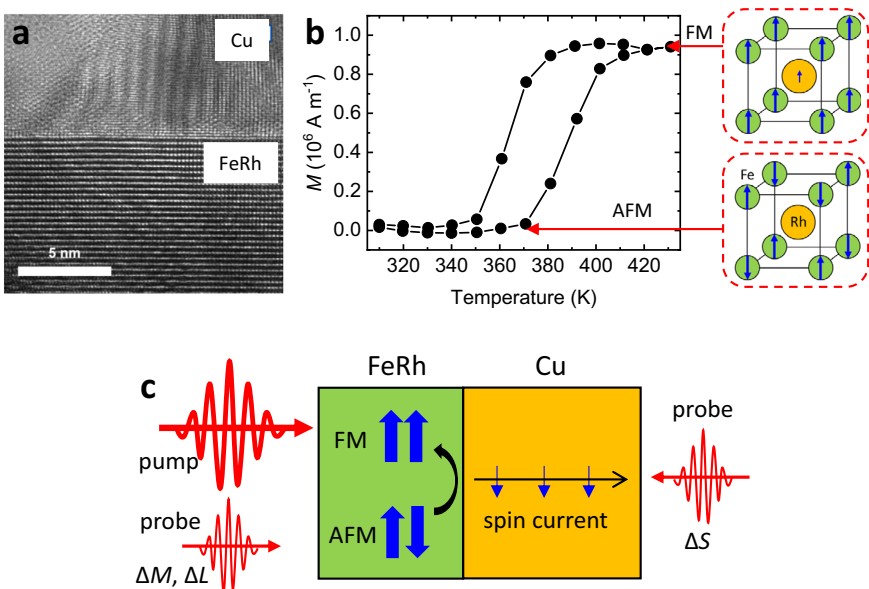

**Fig. 1 | Schematics of the experiment. a** Transmission-electron-microscope image of the FeRh/Cu heterostructure. A white scale bar indicates 5 nm. The FeRh/Cu interface shows a clean and flat morphology. **b** Magnetization versus temperature of FeRh. At temperature less than 350 K, FeRh becomes an antiferromagnetic (AFM) phase with negligible magnetization. At temperature above 400 K, FeRh becomes a ferromagnetic (FM) phase with a net magnetization of ~10$^6$ A m$^{-1}$. The right insets are the configurations of atomic moments (blue arrows) of the AFM and FM phases of FeRh. **c** Schematics of the ultrafast-magnetization-driven spin current. The pump pulse triggers the phase transition from the AFM phase to FM phase of FeRh, which induced a spin current to Cu, where the blue arrow indicates the spin polarization, the black arrow indicates the flow of spin. The probe pulse detects the magnetization ($\Delta M$), lattice expansion ($\Delta L$) of FeRh, and spin accumulation ($\Delta S$) of Cu.

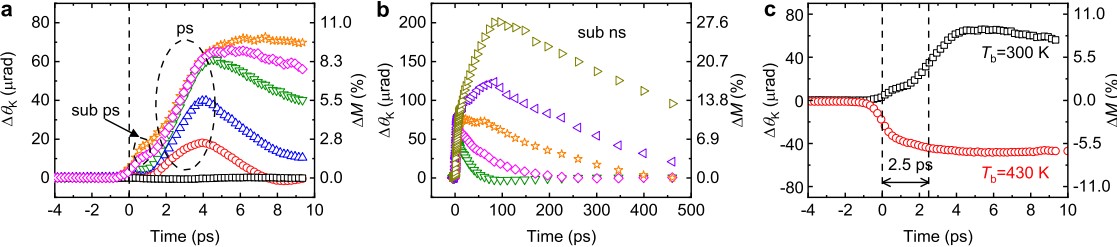

**Fig. 2 | Ultrafast magnetization of FeRh in FeRh/Cu heterostructure.** The dynamic Kerr rotation ($\Delta\theta_K$) by ultrafast magnetization of FeRh at (**a**) short time-scale <10 ps (sub-ps and ps dynamics) and (**b**) long timescale <500 ps (sub-ns dynamics). The magnetization direction of the FM phase of FeRh is set by an external magnetic field of 0.15 T, which is along the out-of-plane direction. The left $y$-axis is the dynamic Kerr rotation, and the right $y$-axis is the relative magnetization ($\Delta M$). $\Delta M$ is determined as $\Delta M = \frac{\Delta\theta_K}{\theta_K} \div \frac{M_z}{M_s}$, where $\theta_K$ is the static Kerr rotation of 5.8 mrad for the saturation magnetization ($M_s$) of the FM phase of FeRh, and $M_z$ is the $z$-component of magnetization. With an external field of 0.15 T, $M_z/M_s$ is 0.13. The color indicates the pump fluence in a unit of J m$^{-2}$: 0.9 (black square), 2.2 (red

circles), 3.5 (blue up-triangles), 5.3 (green down-triangles), 7.1 (magenta diamonds), 8.8 (orange stars), 10.6 (purple left-triangles), and 16.4 (dark yellow right-triangles). In **a**, a major response occurs at 2-4 ps (ps dynamics). In addition, a subtle peak appears at 0.5 ps (sub-ps dynamics). A dashed straight line at the time zero indicate the position of the pump pulse. **c** The dynamic Kerr rotation by the ultrafast demagnetization of the FM phase of FeRh with a pump fluence of 7.1 J m$^{-2}$ at a base temperature ($T_b$) of 430 K (red circles). As a reference, ultrafast magnetization at a base temperature of 300 K with the same pump fluence is shown as black squares. There is 2.5 ps time delay between the ultrafast demagnetization and ultrafast magnetization.

with a latent heat. The fast rise of $\Delta M$ at 2-4 ps (ps dynamics) saturates with a pump fluence >5 J m$^{-2}$, whereas the slow rise $\Delta M$ at a longer timescale of ~100 ps (sub-ns dynamics) exhibits further increase with increasing the pump fluence (Fig. 2b). Both ps and sub-ns dynamics during the FeRh phase transition have been observed previously, and the former and latter were attributed to the FM domain nucleation and growth (or coalescence), respectively[33–36]. Although the complete growth of the FM domain takes a long timescale because of a slow domain wall motion, we expect that the magnetic moment of the initially nucleated FM domain is at least partially aligned along the magnetic field, which produces the Zeeman energy term on the magnetic moment, without a time delay from the onset of the domain nucleation. In this study, we focus on the ps dynamics rather than the sub-ns dynamics because the spin current generation is mostly driven by the ps dynamics (shown later). According to the d$M$/d$t$ model[10], the faster magnetization changes, the larger spin current is generated.

In addition to two major dynamics at ps and sub-ns timescales, a subtle peak appears at ~0.5 ps (sub-ps dynamics). Considering such a short timescale, we expect that the sub-ps dynamics could be originated from the modification in the electronic band structure, whose timescale was reported to be 0.35 ps using a photoelectron spectroscopy[37]. However, we find that the sub-ps dynamics of FeRh does not contribute to the spin current generation as shown later.

To emphasize the timescale of the ps dynamics during the phase transition, we compare the phase-transition-driven dynamics to ultrafast-demagnetization-driven one. Whereas the phase transition induces a net magnetization to emerge starting from zero, ultrafast demagnetization reduces the magnetization starting from a finite value. Upon increasing the base temperature to 430 K, the initial phase of FeRh becomes a FM phase (Fig. 1b and Supplementary Section 2). A sudden heating of the FM phase by a pump pulse leads to ultrafast demagnetization within 1 ps (Fig. 2c). The thermalization process among the electron, magnon, and phonon baths can explain this timescale (Supplementary Sections 4 and 5). Importantly, when we compare the decrease of $\Delta M$ by ultrafast demagnetization and the rise of $\Delta M$ by ultrafast magnetization, the timescale of the ultrafast magnetization is delayed by 2.5 ps. Such a delay suggests that the phase transition requires an additional process other than the thermalization.

### Lattice expansion of FeRh
We argue that the 2.5 ps delay originates from the timescale of the lattice expansion. We compare timescales of the thermalization and lattice expansion through the reflectivity change ($\Delta R$) measurement. It

is well known that temperature or strain on lattice causes $\Delta R$. For the pump-probe experiment with conventional metals, temperature rise generates a peak in $\Delta R$ at the electron-phonon thermalization time (temperature-induced $\Delta R$: $\Delta R_T$), and acoustic-wave-induced strain generates a peak in $\Delta R$ at round-trip time of acoustic wave through the metal thickness (acoustic-wave-induce $\Delta R$: $\Delta R_A$)[42]. The lattice expansion during the phase transition also induces strain, and therefore it can produce a peak in $\Delta R$ (lattice-expansion-induced $\Delta R$: $\Delta R_L$). Although both $\Delta R_A$ and $\Delta R_L$ are caused by strain, they can be separated by using their different dependences on the magnetic field. Whereas the acoustic wave does not depend on the magnetic field, the lattice-expansion depends on the phase transition and thus on the magnetic field[43]. To extract the magnetic-field-dependent part, we take the difference of $\Delta R$ without and with magnetic field of 0.15 T. Note that this magnetic-field-dependent part corresponds to a partial change of $\Delta R_L$ by 0.15 T, but it can effectively exclude $\Delta R_T$ and $\Delta R_A$. Indeed, the raw $\Delta R$ shows clear contributions of $\Delta R_T$ and $\Delta R_A$: temperature rise at near 0 ps and an acoustic echo at 5 ps (Fig. 3a). The position of the acoustic echo is determined as $2(d_{FeRh} - d_{surf})/v_s$, where $v_s$ is the sound velo-city of ~5 km s$^{-1}$ of FeRh[44], $d_{FeRh}$ is the thickness of FeRh, and $d_{surf}$ is the surface depth of initial heating by light penetration (assuming a fixed $v_s$ of 5 km s$^{-1}$, $d_{surf}$ of 7 nm can explain the acoustic echo at 5 ps). On the other hand, the magnetic-field-dependent part of $\Delta R$ shows a domi-nant contribution of $\Delta R_L$ at 2.5 ps, which matches the one-way trip of acoustic wave through the FeRh thickness. Note that the $\Delta R_L$ signal at 2.5 ps has a threshold behavior that is a characteristic feature of the phase transition. The same time delay of 2.5 ps in $\Delta M$ (emergence of the FM phase) and $\Delta L$ (lattice expansion) suggests that the timescale of the domain nucleation during the phase transition is limited by the speed of the lattice expansion.

The causal relation between $\Delta M$ and $\Delta L$ is a chicken-and-egg problem for the mechanism of the phase transition in FeRh. We argue that the speed limit of the phase transition originates from the struc-tural dynamics, speed of acoustic wave. However, if one can find an explanation for 2.5 ps in terms of the magnetic dynamics, such as speeds of precession or domain wall motion, the speed limit could be related to the magnetic origin.

### Spin accumulation on Cu
Next, we measure the spin accumulation on Cu driven by ultrafast magnetization of FeRh in the FeRh/Cu heterostructure. When a pump pulse triggers the phase transition of FeRh, a probe pulse detects spin accumulation ($\Delta S$) in the conduction electron bath of Cu. For NM materials with no magnon bath, only the spin

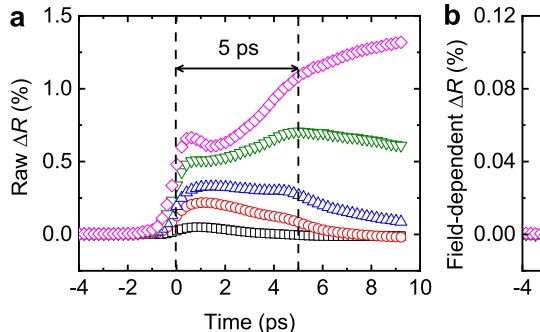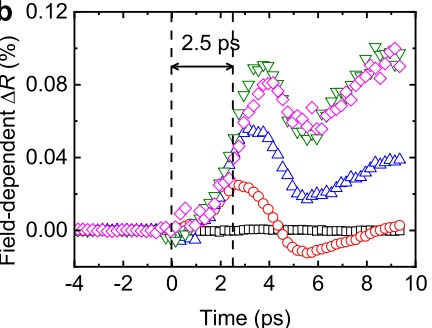

**Fig. 3 | Lattice expansion of FeRh in FeRh/Cu heterostructure. a, b** The reflectivity change ($\Delta R$) during the phase transition of FeRh. The color indicates the pump fluence in a unit of J m$^{-2}$: 0.9 (black square), 2.2 (red circles), 3.5 (blue up-triangles), 5.3 (green down-triangles), and 7.1 (magenta diamonds). **a** The raw $\Delta R$ consists of the temperature-induced ($\Delta R_T$). acoustic-wave-induced ($\Delta R_A$), and lattice-expansion-induced ($\Delta R_L$) ones: temperature rise produces a fast rising in $\Delta R_T$ at around time zero; acoustic wave produces a peak of $\Delta R_A$ at 5 ps, which corresponds to a round trip of acoustic wave through the FeRh thickness; a nonlinear dependence on the pump fluence comes from $\Delta R_L$, which is caused by the phase transition. **b** The magnetic-field-dependent part of $\Delta R$, difference of $\Delta R$ with and without an external magnetic field of 0.15 T, comes from a partial change of $\Delta R_L$ by the magnetic field without a contribution from $\Delta R_T$ and $\Delta R_A$. It shows a significant rising at 2.5 ps, which has a nontrivial dependence on the pump fluence: a threshold at pump fluence of >2 J m$^{-2}$ and a saturation at a pump fluence of >5 J m$^{-2}$.

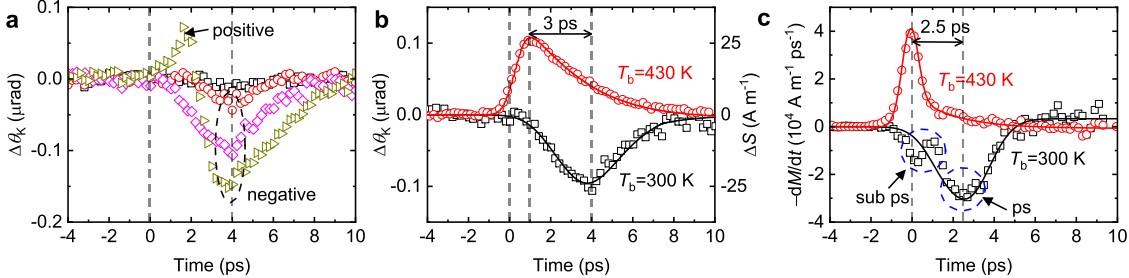

**Fig. 4 | Spin accumulation on Cu in FeRh/Cu heterostructure. a, b** The dynamic Kerr rotation ($\Delta\theta_K$) by the spin accumulation on Cu. **a** The pump fluence dependence of the spin accumulation at a fixed base temperature of 300 K. The color indicates the pump fluence in a unit of J m$^{-2}$: 0.9 (black square), 2.2 (red circles), 7.1 (magenta diamonds), and 28 (dark yellow right-triangles). At a pump fluence >2 J m$^{-2}$, a negative spin accumulation appears at 4 ps. With a very large pump fluence of 28 J m$^{-2}$, a positive spin accumulation appears at 1.5 ps in addition to the negative spin accumulation. **b** The base temperature ($T_b$) dependence of the spin accumulation at a fixed pump fluence of 7.1 J m$^{-2}$. At $T_b$ of 430 K, the initial phase of FeRh becomes ferromagnetic. Then, a positive spin accumulation appears at 1 ps (red circles). As a reference, a negative spin accumulation at $T_b$ of 300 K, with an initial AFM phase of FeRh, is shown as black squares. The black and red solid lines are the results of the spin transport simulation in Fig. 5c, d. The left $y$-axis is the dynamic Kerr rotation, and the right $y$-axis is the spin accumulation ($\Delta S$) on Cu in a unit of magnetization density using a conversion factor[46] of $4 \times 10^{-9}$ rad m A$^{-1}$. **c** The negative time-derivative of magnetization ($-dM/dt$) of FeRh, obtained from the $\Delta M$ results of Fig. 2c. The black squares and red circles are from ultrafast magnetization and ultrafast demagnetization, respectively, at $T_b$ of 300 K and 430 K. The $-dM/dt$ at $T_b$ of 300 K shows the sub-ps and ps dynamics at <1 ps and 2.5 ps, respectively. The $-dM/dt$ at $T_b$ of 430 K shows the b-ps dynamics at <1 ps. The black solid line is a smooth fitting for the ps dynamics of ultrafast magnetization. The red solid line is a smooth fitting for the sub-ps dynamics of ultrafast demagnetization.

polarization of the conduction electrons is responsible for the Kerr rotation. We observe a negative spin polarization on Cu with a peak position at 4 ps, clearly indicating that conduction electrons carry spin current from FeRh to Cu (Fig. 4a). The negative peak at 4 ps is a common feature with pump fluences of >2 J m$^{-2}$, but an additional positive peak occurs at 1.5 ps with a very high pump fluence of 28 J m$^{-2}$. We expect that a high pump fluence induces a significant rise in the steady-state temperature (Supplementary Section 6), which may cause the initial FeRh to have a small portion of the FM phase. For a clear comparison between the initial AFM phase and FM phase, we measure ultrafast-demagnetization-driven spin accumulation with a complete FM phase of FeRh at a base temperature of 430 K (Fig. 4b). Ultrafast demagnetization generates a positive spin polarization on Cu with a peak position at 1 ps, which is 3 ps faster than 4 ps of the negative peak for the ultrafast-magnetization-driven $\Delta S$.

The sign and time delay of $\Delta S$ provide an important clue for the underlying mechanism. The opposite sign of $\Delta S$ driven by the ultrafast magnetization and demagnetization of FeRh disproves any mechanism based on the spin filter effect. The sign of the spin polarization by the spin filter effect is determined by the magnetization direction of the FM phase of FeRh, which is set by the magnetic field, and it is the same for ultrafast magnetization and demagnetization processes. In addition, the 4 ps time delay in $\Delta S$ cannot be explained by the hot electron effect because the transport of hot electrons takes less than 1 ps (Supplementary Section 5). On the other hand, the sign and time delay in $\Delta S$ on Cu has a close relation to $\Delta M$ of FeRh. According to the d$M$/d$t$ model, $-dM/dt$ of the magnon bath acts as spin generation on the electron bath (Fig. 4c). The sign of $\Delta S$ matches well to the sign of $-dM/dt$ of FeRh: ultrafast magnetization/demagnetization of FeRh induces a negative/positive $\Delta S$. In addition, a time shift of 3 ps in $\Delta S$ between the ultrafast magnetization and demagnetization is close to that of 2.5 ps in $-dM/dt$. This result reveals a critical role of the angular momentum transfer between the magnon and electron baths for the spin current generation.

In addition to the ps-dynamics at 2.5 ps, the sub-ps dynamics during the phase transition of FeRh is expected to generate spin current (Fig. 4c). However, we find that the $\Delta S$ of Cu has a close correlation only to the ps-dynamics but not to the sub-ps dynamics (Fig. 4b). Further research is required to understand why the sub-ps dynamics does not contribute to the spin current generation.

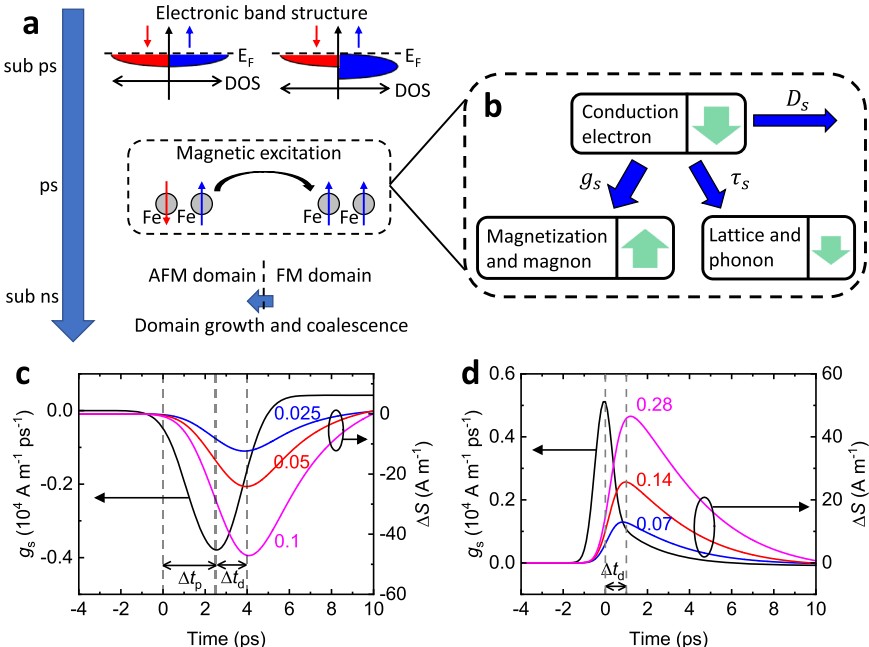

**Fig. 5 | Process of angular momentum transfer during phase transition of FeRh.**
**a** Different timescales during the phase transition of FeRh: a modification of the electronic band structure occurs at sub-ps timescale[37]; a nucleation of ferromagnetic domains, mediated by magnetic excitation, occurs at ps timescale[33–36]; a domain growth and coalescence, mediated by domain wall motion, occurs at sub-ns timescale[33–36]. **b** Angular momentum transfer during the domain nucleation of FeRh. Initially, angular momentum is supplied from the electron bath to the magnon bath with a spin generation rate of $g_s$. Then, the magnon bath has a positive magnetization (green arrows), and the electron bath has a negative magnetization (green arrows). The angular momentum of the electron bath of FeRh can relax to the phonon bath of FeRh with a spin relaxation time of $\tau_s$ and diffuse to the electron bath of adjacent layers with a diffusion constant of $D$. The flow of angular momentum among electron, magnon, and phonon baths are indicated as blue

arrows. The spin transport simulation for the (**c**) ultrafast-magnetization-driven and (**d**) ultrafast-demagnetization-driven spin current. The black lines in **c** and **d** are the spin generation rate on the electron bath of FeRh, determined as $g_s = -\frac{dM_{FeRh}}{dt} \times \frac{M_z}{M_s}$, where $dM/dt$ is obtained from smooth fittings of Fig. 4c, and $M_z/M_s$ of 0.13 with an external magnetic field of 0.15 T. The red, blue, and magenta lines are the simulation results of the spin accumulation ($\Delta S$) on the electron bath of Cu surface. The number indicates $\tau_s$ of the conduction electrons in FeRh: $\tau_s$ of 0.025 ps (**c**) and 0.07 ps (**d**) for the blue lines; $\tau_s$ of 0.05 ps (**c**) and 0.14 ps (**d**) for the red lines; $\tau_s$ of 0.1 ps (**c**) and 0.28 ps (**d**) for the magenta lines. The red lines of **c** and **d** correspond to the solid lines in Fig. 4b. The dashed-vertical lines in **c** and **d** indicate the time delays for $g_s$ and $\Delta S$. In **c**, the peak of $g_s$ has a time delay by phase transition ($\Delta t_p$) of 2.5 ps, and the peak of $\Delta S$ has a time delay by diffusion ($\Delta t_d$) of 1.5 ps in addition to $\Delta t_p$. In **d**, the peak of $g_s$ has no time delay, and the peak of $\Delta S$ has only $\Delta t_d$ of 1 ps.

## Spin transport simulation

For a quantitative analysis of the angular momentum transfer, we perform a simulation of the combined process of spin generation, diffusion, and relaxation in the electron bath (Fig. 5). For the spin generation, we make two assumptions: 1) the ps dynamics of the domain nucleation during the phase transition is mediated by the magnetic excitations, i.e., magnons (Fig. 5a); 2) a change of angular momentum of the magnon bath is quickly supplied by the conduction electron bath (Fig. 5b). Then, $\Delta M$ of the magnon bath is entirely

converted to the spin polarization of the conduction electron bath, and the spin generation rate ($g_s$) of the electron bath is expressed as $g_s = -dM/dt$. The generated spins in the electron bath can diffuse spatially via spin diffusion and dissipate to phonons via spin relaxation (Fig. 5b). The magnitude of the spatial spin current is determined by the competition between the diffusion time and relaxation time. We determine the transport parameters, such as diffusion constants and density of states, from electrical measurements and literature reports (see Method sections and Table I). The simulation result for the spin transport well explains the peak position of $\Delta S$: the negative peak at 4 ps for the ultrafast-magnetization-driven (phase-transition-driven) $\Delta S$ is the combined result of the time scale for the phase transition of FeRh ($\Delta t_p = 2.5$ ps) and that for the diffusive transport from FeRh to Cu ($\Delta t_d = 1.5$ ps) (Fig. 5c), and the positive peak at 1 ps for the ultrafast-demagnetization-driven $\Delta S$ is almost exclusively determined by $\Delta t_d$ (Fig. 5d). Note that $\Delta t_d$ is ~0.5 ps longer in Fig. 5c than in Fig. 5d because the electron mobility of FeRh is smaller in the AFM phase than in the FM phase (Table I).

The only unknown parameter for the spin transport simulation is the spin relaxation time ($\tau_s$). From the magnitude of $\Delta S$, we determine $\tau_s$ of the conduction electrons of FeRh, which describes the speed of angular momentum dissipation from the electron bath to the phonon bath. The longer $\tau_s$ of FeRh is, the more spin is transported from FeRh to Cu before the dissipation to phonons (the long $\tau_s$ of Cu has a negligible effect on the simulation result). Fitting the amplitude of the spin accumulation between the experiment and simulation, we determine $\tau_s$ of 0.05 ps for the AFM phase of FeRh. We also perform a similar

## Table I | Material parameters for spin transport simulation

|  | FM FeRh | AFM FeRh | Cu |
|---|---|---|---|
| $N_F$ ($10^{47}$ J m$^{-3}$) | 8.25 | 3.43 | 1.6 |
| $\sigma$ ($10^6$ Ω$^{-1}$ m$^{-1}$) | 1.3 | 0.8 | 50 |
| $D$ (nm$^2$ ps$^{-1}$) | 62 | 91 | 12200 |
| $\tau_s$ (ps) | 0.14 | 0.05 | 13 |
| $l_s$ (nm) | 2.9 | 2.1 | 400 |

$N_F$ is the electronic density of state at the Fermi level, $\sigma$ is the electrical conductivity, $D$ is the electrical diffusivity, $\tau_s$ is the spin relaxation time, and $l_s$ is the spin relaxation length. $N_F$ values are determined as $N_F = \frac{3\gamma}{\pi^2 k_B^2}$, where $\gamma$ is the coefficient of the electronic heat capacitance from ref. [44], $k_B$ is the Boltzmann constant. We assume that the spin density states ($N_s$) for the spin transport simulation can be approximated to be a half of $N_F$. $\sigma$ values are measured using a four-point probe method. $D$ values are obtained from $\sigma$ values using the relation of $D = \frac{\sigma}{e^2 N_F}$, where $e$ is the elementary charge. $\tau_s$ values of FeRh are determined by comparing the spin accumulation experiment and spin transport simulation (see Methods). $\tau_s$ values of Cu are obtained from $l_s$ of 400 nm from ref. [50]. using the relation of $l_s = \sqrt{D\tau_s}$.

simulation for the ultrafast-demagnetization-driven spin accumulation and determine $\tau_s$ of 0.14 ps for the FM phase of FeRh. The difference in $\tau_s$ between the AFM and FM phases could be due to the different band structures of electrons, magnons, and phonons. Recently, a significant change in the damping constant of FeRh during the phase transition was reported[45]. Assuming that the damping constant and the spin relaxation rate are positively correlated, ref. 45. suggested that the spin lifetime is much shorter in the AFM phase than in the FM phase. $\tau_s$ can be converted to the spin diffusion length, $l_s$, using the relation of $l_s = \sqrt{D\tau_s}$, where $D$ is the electronic diffusivity of FeRh, and we obtain $l_s$ values of 2.1 nm and 2.9 nm, respectively, for the AFM and the FM phases of FeRh, which are smaller than $l_s$ of 7 nm of pure Fe[46]. According to the Elliot-Yafet mechanism[4,47,48], the spin relaxation of the electron bath is due to incoherent electron-phonon scattering in the presence of spin-orbit coupling. We expect that the strong spin-orbit coupling of Rh[49] enhances the spin-flip probability during the electron-phonon scattering.

Our work combines two-seemingly-unrelated phenomena: spin current and phase transition. An integrated understanding of the spatial flow of angular momentum (spin current) and the relocation of angular momentum inside a material (phase transition) will expand the research area of spintronics. In addition, a dynamic coupling between the spin current and phase transition could be useful for the high-speed operation of memory devices, such as magnetic memory and phase change memory.

## Methods

### Film growth

A MgO (001) substrate/$Fe_{50}Rh_{50}$ (20 nm)/Cu (120 nm)/$SiO_2$ (4 nm) stacking structure was fabricated using molecular beam epitaxy (MBE) and sputtering. An epitaxial FeRh layer was grown on a MgO (001) substrate at 450 °C by co-evaporating Fe and Rh from separate sources in an ultrahigh vacuum MBE chamber with a base pressure of ~$10^{-10}$ Torr, followed by post-annealing at 600 °C. To achieve a strong spin current, the thickness of FeRh should be close to the spin diffusion length of FeRh[46]. However, we found that the AFM phase of FeRh becomes incomplete when its thickness becomes too thin. The thickness of FeRh was chosen to 20 nm to have a complete AFM phase at 300 K (Supplementary Sections 1 and 2). A 120-nm-thick Cu layer was then grown on the FeRh layer at room temperature. The thickness of Cu should be much thicker than the light penetration depth but smaller than the spin diffusion length of Cu[50]. At the Cu thickness of 120 nm, the probe on the Cu side only sees the spin accumulation on Cu without any contribution from the magnetization of FeRh[46]. After the growth of FeRh and Cu, the films are immediately transferred to the sputter chamber, and an additional capping layer of a 4-nm-thick $SiO_2$ was grown on top of the Cu layer using RF sputter at Ar pressure of $2 \times 10^{-2}$ Torr at room temperature. The $SiO_2$ capping layer prevents the oxidation of Cu so that our films maintain its property.

### Optical measurement

We used a pump-probe optical technique to observe both the phase transition of FeRh and the spin accumulation on Cu in the time domain. A pulsed laser was generated using a Ti-sapphire oscillator with a repetition rate of 80 MHz and a wavelength of 785 nm. The laser beam was divided into pump and probe beams by a polarizing beam splitter. The time delay between the pump and probe beams was controlled using a motorized mechanical stage. The pump and probe beams were modulated by an electro-optic modulator and optical chopper, respectively, at frequencies of 1 MHz and 200 Hz. The full widths at half maximum of the time correlation of the pump and probe were determined to be 1.2 ps. Considering the large group velocity dispersion of EOM, we expect that FWHM is 1.0 and 0.2 ps for pump and probe, respectively. Both the pump and probe beams were focused to a spot size of 3 μm ($1/e^2$ radius) using a 20× objective lens.

The pump triggers the phase transition of FeRh, and the probe detects magnetization of FeRh or spin accumulation on Cu via magneto-optical Kerr effect (MOKE) and lattice expansion of FeRh via strain-induced reflectivity change. For the MOKE, we used a polar MOKE geometry and aligned the magnetization of FeRh to the out-of-plain direction using a ring magnet, with a magnetic field of 0.15 T. Depending on the position of probe, either on the FeRh side or on the Cu side, we used different optical setup, whose schematics and explanation are shown in Supplementary Section 3.

### Spin transport simulation

To simulate the diffusive transport of the spins of conduction electrons, we used the following equation[46]:

$$\frac{\partial \mu_s}{\partial t} = D \frac{\partial^2 \mu_s}{\partial z^2} - \frac{\mu_s}{\tau_s} + \frac{g_s}{\mu_B N_s}, \tag{1}$$

where $\mu_s$ is the spin chemical potential, $t$ is time, $z$ is the spatial coordinate along the film thickness, $D$ is the diffusion constant, $\tau_s$ is the spin relaxation time, $g_s$ is the spin generation rate, $\mu_B$ is the Bohr magneton, and $N_s$ is the spin density of state. $g_s$ is determined as $g_s = -\frac{dM}{dt}$, where $M$ is obtained from the experimentally measured ultrafast magnetization of FeRh (Fig. 2c). Since the FeRh thickness is thicker than the light penetration depth, we assume a non-uniform $g_s$ along the FeRh thickness, and distribution of $g_s$ is determined from the distribution of the light absorption (Supplementary Note 4). $D$ was determined from the electrical conductivity as $D = \frac{\sigma_e}{e^2 N_F}$, where $\sigma_e$ is the electronic conductivity of the material, $e$ is the elementary charge, and $N_F$ is the electronic density of states at the Fermi level. The $\sigma_e$ values of $1.3 \times 10^6$, $0.8 \times 10^6$, and $50 \times 10^6$ $\Omega^{-1}$ $m^{-1}$ of FM FeRh, AFM FeRh, and Cu, respectively, are measured using a four-point probe method. $N_s$ was determined as $N_s = \frac{N_F}{2}$. $\mu_s$ of FeRh and Cu are connected at the interface with an interfacial spin conductance of $G_s$, which is determined as $G_s = \frac{G_e}{2e^2}$, where $G_e$ is the electrical conductance at the interface and $e$ is the elementary charge. Since $G_e$ value of the FeRh/Cu interface is not known, we used $G_e$ of $2 \times 10^{15}$ $\Omega^{-1}$ $m^{-2}$ of the permalloy/Cu and Co/Cu interfaces[51]. The diffusive spin current was estimated using $J_s = \frac{D}{N_s} \frac{\partial \mu_s}{\partial z}$ at the bulk of FeRh and Cu and $J_s = G_s \Delta \mu_s$ at the FeRh/Cu interface, where $\Delta \mu_s$ is the difference in $\mu_s$ at the interface. Although we used a $G_s$ value, which is not taken from the actual interface of FeRh/Cu, we argue that the uncertainty in $G_s$ is not critical. From the spin transport simulation with the thicknesses of 20 nm for FeRh and 120 nm for Cu, we found that the bulk parameter of $D$ has a dominant effect on the spin transport over the interface parameter of $G_s$. The only free parameter for the spin transport simulation is the spin relaxation time ($\tau_s$) of FeRh, and it was determined from the fitting between the experiment and the simulation. The $\tau_s$ of Cu was determined from the reported spin diffusion length ($l_s$) of 400 nm of Cu[50]. The parameters for the spin diffusion simulation are summarized in Table I.

## Data availability

The dataset of the main figures generated in this study is provided in the Supplementary Information/Source Data file. Source data are provided with this paper.

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

## Acknowledgements

K.K. and and G.-M.C. are supported by Samsung Research Funding & Incubation Center of Samsung Electronics under Project Number SRFC-MA2001-03. H.O. and T.T. are supported in part by JST CREST Grant No. JPMJCR18J1, JSPS KAKENHI Grant No. 21H04614. H.W.L is supported by Samsung Science and Technology Foundation (BA-1501-51). K.-J.L. is

supported by the National Research Foundation of Korea (NRF-2020R1A2C3013302). O. Lee was supported by the NRF of Korea (2020M3F3A2A01081635). Device fabrication was supported in part by Advanced Facility Center for Quantum Technology at Sungkyunkwan university.

## Author contributions

G.-M.C. supervised the study. K.K. performed the measurement and analysis for the phase transition and spin current generation. H.O. and T.T. grew the FeRh/Cu heterostructure films and measured the magnetic properties. D.Y. contributed to the inital growth of FeRh. O.L. provided reference FeRh samples and fabrication procedure. K.-J.L. and H.-W.L. provided a theoretical consideration for angular momentum transfer. All authors discussed the results and wrote the manuscript.

## Competing interests

The authors declare no competing interests.
