## [Peer Review File · Nature Communications]

Reviewers' Comments:

Reviewer #1:

Remarks to the Author:

The manuscript titled "Spin current pumping by ultrafast magnetization of FeRh" by Kang et al reports about novel approach to study ultrafast dynamics of phase transition in FeRh from antiferromagnetic to ferromagnetic phase. The approach combines the principles of spintronics and ultrafast magnetism. More particularly, the absolute majority of previously published works dedicated to the problem of ultrafast kinetics of the phase transition in FeRh employed time-resolved pump-and-probe techniques. Very recently, it was realized that the phase transition is accompanied by spintronic phenomena such as spin pumping (see Wang et al, "Spin pumping during the antiferromagnetic- ferromagnetic phase transition of iron-rhodium", Nature Comm 11, 275 (2020)). In this paper, the authors practically employ the dynamics of spin pumping as a probe of kinetics of the phase transition. Unfortunately, I have not noticed that in the proposed interpretation of the experimental results the authors account for the findings of Wang et al. This is probably the origin of most of weaknesses of this very interesting work. Regarding the weaknesses (see the summary below), I do not think that this paper is suitable for publication in the present form. However, I think that this is a very interesting work, where the authors discuss new experimental results obtained with the help of a novel experimental approach (it has not been applied to the problem of FeRh before). As far as I can see, the manuscript contains a sufficient description to reproduce the results reported in the manuscript. Therefore I think that after a proper revision the work will be noticed and will have an impact on magnetic research, in general, and on the rapidly growing field of ultrafast magnetism, in particular.

My main concerns are related to simulations and corresponding interpretation of the results.

Although I very much appreciate the attempt of the authors to simulate the studied phenomenon, the values of many parameters in the simulations raise serious concerns.

- a) Spin relaxation time τ_s of FeRh. The authors should add more explanation about physical meaning of this parameter. I assume that this is spin relaxation time of conducting s-electrons of Fe. This time is obviously different from spin relaxation of localized spins of d-electrons of Fe as well as spins of Rh.
- b) It is not clear where the electron diffusivity D for FeRh is taken from. Please note that in ferromagnetic and antiferromagnetic phases FeRh has substantially different electric conductivities. I assume, it must have an effect of electronic diffusivity, but the authors seem to think differently. It is not clear why.
- c) In many of the simulations the authors take parameters not for FeRh, but for other magnetic metals. In every of these cases the authors should explain why they think that this is a valid substitute.
- d) Wang et al report about a dramatic decrease of damping of FMR upon the phase transition from the AFM to the FM phase. It looks that this result correlates with the finding of the present manuscript – spin relaxation in the AFM phase is nearly 30 times faster than in the FM phase. I wonder the authors think about such a correlation and can explain it.

Minor issue

The authors assume that the dynamics of reflectivity reveals the dynamics of lattice expansion. Although I agree with the authors that this is a valid assumption, I must note that not all specialists studying FeRh would easily accept this statement. I advise the authors to add an explanation showing that in the case of FeRh reflectivity can be considered as a reliable probe of the lattice dynamics.

Reviewer #2:

Remarks to the Author:

Report on the manuscript "Spin current driven by ultrafast magnetization of FeRh" by K. Kang et al.

The authors find ultrafast generation of spin current induced by a phase transition in FeRh, manifested by the angular momentum flow from the electronic bath of an adjacent Cu layer. The angular momentum transfer between different reservoirs (electron spins, magnons, lattice) is essential for understanding the driving forces of ultrafast magnetization dynamics in different

materials. Hence, the manuscript presents an important insight into the possible sources of angular momentum needed for magnetic transformation of the antiferromagnetic phase to the ferromagnetic one. In this regard, it would benefit the discussion to account for ref. 36 in more detail, which, besides showing the sub-picosecond time scale of the phase transition, deals with the intersite transfer of angular momentum in FeRh, backed by DFT calculations.

In the present paper, the ultrafast magnetization in FeRh is followed using the Kerr effect and reflectivity optical response. At the laser fluence of 7.1 J/m^2 , the time scale of the Kerr and reflectivity responses is very well correlated, at $\sim 2.5 \text{ ps}$. Is this correlation expected? The Kerr signal probes the net magnetization, which has a delayed onset as it requires alignment of magnetization in the nucleated ferromagnetic domains along the applied field.

The authors also suggest that the phase transition time scale is set by the lattice expansion. The argumentation with strain wave propagation is of limited relevance, as the phase transition does not seem complete upon laser excitation with 7.1 J/m^2 . Both reflectivity and Kerr signals further increase with increasing pump fluence.

Furthermore, the spin depletion in Cu is only shown for the laser fluence of 7.1 J/m^2 . Are data for other fluences available?

In general, a number of claims is unclear or is not backed by evidence or references, e.g., "only the fast dynamics plays a role in the spin current generation."

"When we inject a pump pulse on FeRh through the MgO substrate, a >local heating< triggers the phase transition of FeRh on a timescale of a few picoseconds."

"...it (the optical probe) also detects the lattice expansion (ΔL) of FeRh via the strain-induced reflectivity change (Fig. 1c)." Explanation of the strain effect or a reference to the existing literature is missing.

Finally, the magnetic characterization of FeRh thin films in Suppl. Fig. 1b-1c shows an approximately 20% lower magnetization than expected for the fully ferromagnetic phase. Was an external magnetic field applied during the temperature loop? The authors might present a magnetic field induced hysteresis loop at 300 K to support their claim of negligible magnetization in the antiferromagnetic phase.

In conclusion, although the observation of the angular momentum transfer from Cu upon the phase transition in FeRh is intriguing, the discussion of this effect is not sufficiently deep, at times lacking context and references for the claims, and contains several inconsistencies which render the work inappropriate for publication in Nature Communications.

Response letter to Referee's comments

Manuscript NCOMMS-22-36165-T

“Spin current driven by ultrafast magnetization of FeRh” by Kyuhwe Kang *et al.*

We thank both referees for their constructive comments. We believe our manuscript is substantially improved as a result. Below, we summarize the major changes and provide point-by-point responses to the referee's comments and suggestions that requires responses. The point-by-point responses are marked in blue. The corresponding corrections are incorporated in the revised manuscript.

Summary of major changes

1. Reviewer 1 questioned the validity of the parameters used in our simulation. We pointed out that important parameters for the spin transport simulation such as electrical conductivity and density of states are determined from material properties of antiferromagnetic and ferromagnetic phases of FeRh. All parameter values are summarized in Table I of the revised manuscript. We also described details of the parameter value determination in the method section.
2. Reviewer 1 requested to consider a previous work [Wang *et al. Nat. Commun.* 11, 275 (2020)] that reports the dramatic decrease of damping constant upon the phase transition from the AFM to the FM phase of FeRh. The damping decrease is interpreted as the spin relaxation time enhancement upon the transition. Our analysis indeed shows that the spin relaxation is about three times longer in the FM phase than in the AFM phase. We included this explanation in the revised manuscript.
3. Reviewer 2 requested to consider a previous [Pressacco *et al. Nat. Commun.* 12, 5088 (2021)] that reports the sub-ps modification of the electronic band structure during the phase transition of FeRh. In our measurement, a significant rise of magnetization occurs at 2~4 ps (ps dynamics). But, in addition to the ps dynamics, we also observed a subtle peak at 0.5 ps (sub-ps dynamics), which may be attributed to the electronic band structure modification. In our revised manuscript, we highlighted both sub-ps and ps dynamics in Figure 2a, 4c, and 5a so that the reviewer can recognize this point. However, we find that the sub-ps dynamics does not contribute to the spin current generation. We included this discussion in the revised manuscript and suggested further research to understand the exact role of the sub-ps dynamics.

4. Both reviewers 1 and 2 requested elaborated explanation for the detection of lattice expansion. We cited a reference to show that acoustic-wave-induced strain can produce a peak in reflectivity (ΔR) [Norris *et al. Rev. Sci. Ins.* 74, 400 (2003)]. Since the lattice expansion induces strain, the lattice expansion can also change ΔR . We showed that the acoustic-wave-induced ΔR and lattice-expansion-induced ΔR can be separated from the magnetic field dependence of ΔR . We included the field-independent and dependent parts of ΔR in Fig. 3 of the revised manuscript.

Reviewer #1 (Remarks to the Author):

The manuscript titled “Spin current pumping by ultrafast magnetization of FeRh” by Kang *et al* reports about novel approach to study ultrafast dynamics of phase transition in FeRh from antiferromagnetic to ferromagnetic phase. The approach combines the principles of spintronics and ultrafast magnetism. More particularly, the absolute majority of previously published works dedicated to the problem of ultrafast kinetics of the phase transition in FeRh employed time-resolved pump-and-probe techniques. Very recently, it was realized that the phase transition is accompanied by spintronic phenomena such as spin pumping (see Wang *et al*, “Spin pumping during the antiferromagnetic–ferromagnetic phase transition of iron–rhodium”, *Nature Comm* 11, 275 (2020)). In this paper, the authors practically employ the dynamics of spin pumping as a probe of kinetics of the phase transition. Unfortunately, I have not noticed that in the proposed interpretation of the experimental results the authors account for the findings of Wang *et al*. This is probably the origin of most of weaknesses of this very interesting work. Regarding the weaknesses (see the summary below), I do not think that this paper is suitable for publication in the present form. However, I think that this is a very interesting work, where the authors discuss new experimental results obtained with the help of a novel experimental approach (it has not been applied to the problem of FeRh before). As far as I can see, the manuscript contains a sufficient description to reproduce the results reported in the manuscript. Therefore I think that after a proper revision the work will be noticed and will have an impact on magnetic research, in general, and on the rapidly growing field of ultrafast magnetism, in particular. My main concerns are related to simulations and corresponding interpretation of the results. Although I very much appreciate the attempt of the authors to simulate the studied phenomenon, the values of many parameters in the simulations raise serious concerns.

a) Spin relaxation time τ_s (τ_s) of FeRh. The authors should add more explanation about physical meaning of this parameter. I assume that this is spin relaxation time of conducting s-electrons of Fe. This time is obviously different from spin relaxation of localized spins of d-electrons of Fe as well as spins of Rh.

(Response) τ_s in our simulation (Figure 5 of the revised manuscript) is the spin relaxation time of conduction electrons (s-electron) transported from FeRh to Cu. We expect that the spin moment of s-electrons (conduction electrons) can be transported from FeRh to Cu, whereas the spin moments of d-electrons are localized on FeRh. Therefore, the MOKE on the Cu side only detects the spin moment of the s-electrons, and τ_s in our simulation is the spin relaxation time of s-electrons. We clarified this point in the lines 160~163 and 214~216 of the revised manuscript.

b) It is not clear where the electron diffusivity D for FeRh is taken from. Please note that in ferromagnetic and antiferromagnetic phases FeRh has substantially different electric conductivities. I assume, it must have an effect of electronic diffusivity, but the authors seem to think differently. It is not clear why.

(Response) The electrical diffusivities (D) of FM and AFM phases of FeRh and Cu are obtained from the electrical conductivities (σ_e) using the relation of $D = \frac{\sigma_e}{e^2 N_F}$, where σ_e is the electrical conductivity, e is the elementary charge, and N_F is the electronic density of states at the Fermi level. The σ_e values of 1.3×10^6 , 0.8×10^6 , and $50 \times 10^6 \Omega^{-1} \text{ m}^{-1}$ of FM FeRh, AFM FeRh, and Cu, respectively, are measured using a four-point probe method. N_F values of FM FeRh, AFM FeRh, and Cu are determined as $N_F = \frac{3\gamma}{\pi^2 k_B^2}$, where γ is the coefficient of the electronic heat capacitance [Phys. Rev. Lett. 109, 255901 (2012)], and k_B is the Boltzmann constant. We included this explanation in the method section and summarized all parameters for the spin transport simulation in Table I of the revised manuscript.

	FM FeRh	AFM FeRh	Cu
N_F (10^{47} J m^{-3})	8.25	3.43	1.6
σ ($10^6 \Omega^{-1} \text{ m}^{-1}$)	1.3	0.8	50
D ($\text{nm}^2 \text{ ps}^{-1}$)	62	91	12200
τ_s (ps)	0.14	0.05	13
l_s (nm)	2.9	2.1	400

Table I. Material properties for spin transport simulation. N_F is the electronic density of state at the Fermi level, σ is the electrical conductivity, D is the electrical diffusivity, τ_s is the spin relaxation time, and l_s is the spin relaxation length. N_F values are determined as $N_F = \frac{3\gamma}{\pi^2 k_B^2}$, where γ is the coefficient of the electronic heat capacitance from Ref. 43, k_B is the Boltzmann constant. We assume that the spin density of states (N_s) for the spin diffusion simulation can be approximated to be a half of N_F . σ values are measured using a four-point probe method. D values are obtained from σ values using the relation of $D = \frac{\sigma}{e^2 N_F}$, where e is the elementary charge. τ_s values of FeRh are determined by comparing the spin accumulation experiment and spin transport simulation (see Methods). τ_s values of Cu are obtained from l_s of 400 nm from Ref. 49 using the relation of $l_s = \sqrt{D\tau_s}$.

In addition to the bulk parameter of D , we also need to know the interface parameter of G_s , the spin conductance at the FeRh/Cu interface. Since G_s value of the FeRh/Cu interface is not known, we used a value for the permalloy/Cu and Co/Cu interfaces. However, we argue that the uncertainty in G_s is not that critical. From the spin transport simulation with the thicknesses of 20 nm for FeRh and 120 nm for Cu, we found that that the bulk parameter of D of FeRh and Cu has a dominant effect on the spin transport over the interface parameter of G_s of FeRh/Cu interface. We included this explanation in the method section of the revised manuscript.

c) In many of the simulations the authors take parameters not for FeRh, but for other magnetic metals. In every of these cases the authors should explain why they think that this is a valid substitute.

(Response) For the spin transport simulation (Figure 5 of the revised manuscript), which is a key analysis in our modeling, we used parameters of FM and AFM phases of FeRh, such as electrical conductivities and density of states. For the thermalization simulation (Supplementary Figure 5), we need to know additional parameters of electron-phonon coupling, electron-magnon coupling, and magnon-phonon coupling. These parameters of FeRh unknown, so we used typical values of conventional ferromagnetic materials. Despite the uncertainties in these coupling parameters, the thermal simulation gives a rough estimation of the temperature dynamics. We included this explanation in the Supplementary Section 5.

d) Wang et al report about a dramatic decrease of damping of FMR upon the phase transition from the AFM to the FM phase. It looks that this result correlates with the finding of the present manuscript – spin relaxation in the AFM phase is nearly 30 times faster than in the FM phase. I wonder the authors think about such a correlation and can explain it.

(Response) We determined the spin relaxation time (τ_s) of the AFM and FM phase from the quantitative comparison between the magnetization change (ΔM) on the FeRh side and the spin accumulation (ΔS) on the Cu side. We found that τ_s of 0.05 ps of the AFM phase is about three times smaller than that of 0.14 ps of the FM phase. Assuming that the damping constant and the spin relaxation rate are positively correlated, the trend of τ_s increase upon the phase transition is consistent with Wang's report. We included this explanation on the line 222~225 of the revised manuscript.

Minor issue

The authors assume that the dynamics of reflectivity reveals the dynamics of lattice expansion. Although I agree with the authors that this is a valid assumption, I must note that not all specialists studying FeRh would easily accept this statement. I advise the authors to add an explanation showing that in the case of FeRh reflectivity can be considered as a reliable probe of the lattice dynamics.

(Response) It is well known that temperature or strain on a lattice changes the reflectivity (ΔR) of light. For the pump-probe experiment with conventional metals, temperature rise generates a peak in ΔR at the electron-phonon thermalization time (temperature-induced ΔR : ΔR_T), and acoustic-wave-induced strain generates a peak in ΔR at round-trip time of acoustic wave through the metal thickness (acoustic-wave-induced ΔR : ΔR_A) [*Rev. Sci. Ins.* **74**, 400 (2003)]. The lattice expansion during the phase transition also induces strain, and therefore it can produce a peak in ΔR (lattice-expansion-induced ΔR : ΔR_L). Although both ΔR_A and ΔR_L are caused by strain, they can be separated from the magnetic field dependence (Fig. R1). Whereas the acoustic wave does not depend on the magnetic field, the lattice-expansion depends on the phase transition and thus on the magnetic field [*Phys. Rev. B* **72**, 214432 (2005)]. Therefore, we assume that the field-dependent part of ΔR corresponds to ΔR_L . The measured ΔR_L shows a threshold behavior that is a characteristic feature of the phase transition. We included this explanation in the lines 137~157 of the revised manuscript. We also included both the magnetic field independent and dependent parts of ΔR in Figure 3 of the revised manuscript.

Figure R1. Lattice expansion of FeRh in FeRh/Cu heterostructure. **a,b,**The reflectivity change (ΔR) during the phase transition of FeRh. The color indicates the pump fluence in a unit of J m^{-2} : 0.9 (black square), 2.2 (red circles), 3.5 (blue up-triangles), 5.3 (green down-triangles), and 7.1 (magenta diamonds). Comparing ΔR with and without an external magnetic field of 0.15 T, we separate the magnetic-field independent (a) and dependent (b) parts of ΔR . **a,** The magnetic-field independent ΔR consists of the temperature-induced change (ΔR_T) and acoustic-wave-induced change (ΔR_A): temperature rise produces a fast rising in ΔR at around time zero, and acoustic wave produces an acoustic echo at 5 ps, which corresponds to a round trip of an acoustic wave through the FeRh thickness. **b,** the magnetic-field dependent ΔR comes from the lattice expansion (ΔR_L) during the phase transition. At 2.5 ps, ΔR_L shows a significant rising, which has a nontrivial dependence on the pump fluence: a threshold at pump fluence of $>2 \text{ J m}^{-2}$ and a saturation at a pump fluence of $>5 \text{ J m}^{-2}$.

Reviewer #2 (Remarks to the Author):

Report on the manuscript “Spin current driven by ultrafast magnetization of FeRh” by K. Kang et al.

The authors find ultrafast generation of spin current induced by a phase transition in FeRh, manifested by the angular momentum flow from the electronic bath of an adjacent Cu layer. The angular momentum transfer between different reservoirs (electron spins, magnons, lattice) is essential for understanding the driving forces of ultrafast magnetization dynamics in different materials. Hence, the manuscript presents an important insight into the possible sources of angular momentum needed for magnetic transformation of the antiferromagnetic phase to the ferromagnetic one. In this regard, it would benefit the discussion to account for ref. 36 in more detail, which, besides showing the sub-picosecond time scale of the phase transition, deals with the intersite transfer of angular momentum in FeRh, backed by DFT calculations.

(Response) In addition to two major dynamics at ps and sub-ns timescales, we also observed a subtle peak in the Kerr signal at around 0.5 ps (sub-ps dynamics in Fig. R2). Considering such a fast timescale, we expect that this sub-ps dynamics may be related to the modification in the electronic band structure [Pressacco et al. Nat. Commun. 12, 5088 (2021)].

Figure R2. Ultrafast magnetization of FeRh in FeRh/Cu heterostructure. a,b, The dynamic Kerr rotation ($\Delta\theta_K$) by ultrafast magnetization of FeRh at (a) short timescale <10 ps and (b) long timescale <500 ps. The magnetization direction of the FM phase of FeRh is set by an external magnetic field of ± 0.15 T, which is along the out-of-plane direction. The left y-axis is the dynamic Kerr rotation, and the right y-axis is the relative magnetization (ΔM). ΔM is determined as $\Delta M = \frac{\Delta\theta_K}{\theta_K} \div \frac{M_z}{M_s}$, where θ_K is the static Kerr rotation of 5.8 mrad for the saturation magnetization (M_s) of the FM phase of FeRh, and M_z is the z-component of magnetization. With an external field of 0.15 T, M_z/M_s is 0.13. The color indicates the pump fluence in a unit of J m^{-2} : 0.9 (black square), 2.2 (red circles), 3.5 (blue up-triangles), 5.3 (green down-triangles), 7.1 (magna diamonds), 8.8 (orange stars), 10.6 (purple left-triangles), and 16.4 (dark yellow right-triangles). In (a), a major response occurs at 2~4 ps (ps dynamics). In addition, a subtle peak appears at 0.5 ps (sub-ps dynamics). A dashed straight line at the time zero indicate the position of the pump pulse.

According to the dM/dt model, both sup-ps and ps dynamics of FeRh can have significant contributions to the spin current generation, and the time derivative of the measured magnetization change of FeRh indeed predicts two components at sub-ps and ps timescales (Fig. R3b). However, we found that the spin accumulation on Cu is explained only by the ps dynamics not by the sup-ps dynamics of FeRh (Fig. R3a). At this moment, we do not understand why the sup-ps dynamics does not contribute to the spin current generation. Further research is required to solve this issue. We included this explanation on the line 118~123 and 188~192 of the revised manuscript. We also highlighted both sub-ps and ps dynamics of FeRh in Figs. 2a, 4c, and 5a in the revised manuscript.

Figure R3. Spin accumulation on Cu in FeRh/Cu heterostructure. **a**, The base temperature (T_b) dependence of the spin accumulation at a fixed pump fluence of 7.1 J m^{-2} . At T_b of 430 K, the initial phase of FeRh becomes ferromagnetic. Then, a positive spin accumulation appears at 1 ps (red circles). As a reference, a negative spin accumulation at T_b of 300 K, with an initial AFM phase of FeRh, is shown as black squares. The black and red solid lines are the results of the spin transport simulation in Figs. 5c and 5d. The left y-axis is the dynamic Kerr rotation, and the right y-axis is the spin accumulation (ΔS) on Cu in a unit of magnetization density using a conversion factor of $4 \times 10^{-9} \text{ rad m A}^{-1}$. **b**, The negative time-derivative of magnetization ($-dM/dt$) of FeRh, obtained from the ΔM results of Fig. 2c. The black squares and red circles are from ultrafast magnetization and ultrafast demagnetization, respectively, at T_b of 300 K and 430 K. The $-dM/dt$ at T_b of 300 K shows the sub-ps and ps dynamics at $<1 \text{ ps}$ and 2.5 ps , respectively. The $-dM/dt$ at T_b of 430 K shows the sub-ps dynamics at $<1 \text{ ps}$. The black solid line is a smooth fitting for the ps dynamics of ultrafast magnetization. The red solid line is a smooth fitting for the sub-ps dynamics of ultrafast demagnetization.

In the present paper, the ultrafast magnetization in FeRh is followed using the Kerr effect and reflectivity optical response. At the laser fluence of 7.1 J/m^2 , the time scale of the Kerr and reflectivity responses is very well correlated, at $\sim 2.5 \text{ ps}$. Is this correlation expected? The Kerr signal probes the net magnetization, which has a delayed onset as it requires alignment of magnetization in the nucleated ferromagnetic domains along the applied field.

(Response) According to the previous works [PRB 73, 060407(R) (2006), PRB 81, 104415 (2010), PRL 108, 087201 (2012)], two different timescales have been observed during the phase transition of FeRh. In our experiment, we also found that the major evolution of magnetization during the phase transition consists of two different timescales: ps dynamics and sub-ps dynamics. Following the previous interpretation, we interpret that the ps dynamics comes from the domain nucleation and the sub-ps dynamics comes from the domain growth. Although the complete growth of the ferromagnetic domain takes a long timescale because of

a slow domain wall motion, we expect that the magnetic moment of the nucleated domain is at least partially aligned along the magnetic field without a time delay from the onset of the domain nucleation, which is ~ 2.5 ps, because of the Zeeman energy term. We included this explanation in the lines 110~114 of the revised manuscript.

This argument is further supported by two considerations. First, the dM/dt modeling (simulation result of Figure 5) well explains both demagnetization-driven and magnetization-driven (phase-transition-driven) spin current. The demagnetization of ferromagnetic phase is caused by the reduction of magnetization rather than by the misalignment of magnetization. Second, the alignment of pre-existing magnetization along the magnetic field proceeds by multiple precessions and damping process, which takes more than a few hundreds ps. Based on these considerations, we argue that 2.5 ps is the timescale for the domain nucleation rather than the timescale for magnetization alignment.

The authors also suggest that the phase transition time scale is set by the lattice expansion. The argumentation with strain wave propagation is of limited relevance, as the phase transition does not seem complete upon laser excitation with 7.1 J/m^2 . Both reflectivity and Kerr signals further increase with increasing pump fluence.

(Response) We measured the dynamics of the phase transition by the reflectivity change (ΔR) and Kerr rotation ($\Delta\theta$). We found that the lattice-expansion-induced ΔR (ΔR_L) and $\Delta\theta$ of FeRh at the ps timescales have three common features: 2.5 ps time delay; threshold at a pump fluence at $>2 \text{ J/m}^2$; saturation at a pump fluence at $>5 \text{ J/m}^2$. From these features, we concluded that the 2.5 ps indicates the onset of the domain nucleation. As the reviewer pointed out, the saturation behavior in the Kerr signal is limited to the ps timescale, and further increases occur at the sub-ns timescale. However, according to the previous works [PRB 73, 060407(R) (2006), PRB 81, 104415 (2010), PRL 108, 087201 (2012)], the sub-ns timescale is related to the domain growth process.

The relevance between the 2.5 ps and acoustic wave propagation is based on our observation of the acoustic-wave-induced induced ΔR (ΔR_A). The ΔR_A and ΔR_L can be separated from the magnetic field dependence of ΔR . We found that a peak in ΔR_A occurs at 5 ps (Fig. R1), and this timescale is determined by the round trip of the acoustic wave along the FeRh thickness. Then, the 2.5 ps in ΔR_L matches the one-way trip of the acoustic wave along the FeRh thickness.

Furthermore, the spin depletion in Cu is only shown for the laser fluence of 7.1 J/m^2 . Are data

for other fluences available?

(Response) We investigated the pump fluence dependence of the spin accumulation on Cu. The threshold and saturation behaviors are observed (Fig. R4). The negative spin accumulation at 4 ps is a common feature with pump fluences of $>2 \text{ J m}^{-2}$, but an additional positive peak occurs at 1.5 ps with a very high pump fluence of 28 J m^{-2} . We included this result in Figure 4 of the revised manuscript.

Figure R4. Spin accumulation on Cu in FeRh/Cu heterostructure. the pump fluence dependence of the spin accumulation at a fixed base temperature of 300 K. The color indicates the pump fluence in a unit of J m^{-2} : 0.9 (black square), 2.2 (red circles), 7.1 (magenta diamonds), and 28 (dark yellow right-triangles). At a pump fluence $>2 \text{ J m}^{-2}$, a negative spin accumulation appears at 4 ps. With a very large pump fluence of 28 J m^{-2} , a positive spin accumulation appears at 1.5 ps in addition to the negative spin accumulation

For the mechanism of this positive peak, we expect that a large pump fluence induces a significant rise in the steady-state temperature (ΔT_{steady}) of FeRh. The ΔT_{steady} by the pump fluence is estimated as [Rev. Sci. Ins. 75, 5119 (2004)],

$$\Delta T_{\text{steady}} = \frac{P_{\text{in}} a_{\text{FeRh}}}{2\sqrt{\pi} w_0 \Lambda},$$

where P_{in} is the incident power of pump light, a_{FeRh} is the light absorption efficiency by the FeRh films, w_0 is the radius of pump light, and Λ is the thermal conductivity of MgO substrate. P_{in} is related to the incident pump fluence (F_{in}), energy per area of a single pump pulse, as

$$F_{\text{in}} = \frac{P_{\text{in}}}{\pi w_0^2} \times \frac{2}{f_{\text{EOM}}},$$

where f_{EOM} is the modulation frequency of electro-optic modulator. Using F_{in} of 28 J/m^2 , f_{EOM}

of 1 MHz, w_0 of 3 μm , a_{FeRh} of 0.4, and Λ of 30 W/mK, we estimate ΔT_{steady} of ~ 40 K. Then, the initial status of FeRh may consist of a small portion of the ferromagnetic phase, which can produce demagnetization-driven spin current. We included this explanation in the Supplementary Note 6.

In general, a number of claims is unclear or is not backed by evidence or references, e.g., “only the fast dynamics plays a role in the spin current generation.”

(Response) We change this sentence as the following.

“In this study, we focus on the ps dynamics rather than the sub-ns dynamics because the spin current generation is mostly driven by the ps dynamics (shown later). According to the dM/dt model, the faster magnetization changes, the larger spin current is generated.”

"When we inject a pump pulse on FeRh through the MgO substrate, a >local heating< triggers the phase transition of FeRh on a timescale of a few picoseconds."

(Response) We remove the expression of “local heating triggers the phase transition” and changed this sentence as the following.

“When we inject a pump pulse on FeRh through the MgO substrate, it triggers the phase transition of FeRh.”

"...it (the optical probe) also detects the lattice expansion (ΔL) of FeRh via the strain-induced reflectivity change (Fig. 1c)." Explanation of the strain effect or a reference to the existing literature is missing.

(Response) It is well known that temperature or strain on a lattice changes the reflectivity (ΔR) of light. For the pump-probe experiment with conventional metals, temperature rise generates a peak in ΔR at the electron-phonon thermalization time (temperature-induced ΔR : ΔR_T), and acoustic-wave-induced strain generates a peak in ΔR at round-trip time of acoustic wave through the metal thickness (acoustic-wave-induced ΔR : ΔR_A) [*Rev. Sci. Ins.* **74**, 400 (2003)]. The lattice expansion during the phase transition also induces strain, and therefore it can produce a peak in ΔR (lattice-expansion-induced ΔR : ΔR_L). Although both ΔR_A and ΔR_L are caused by strain, they can be separated from the magnetic field dependence (Fig. R1). Whereas the acoustic wave does not depend on the magnetic field, the lattice-expansion depends on the phase transition and thus on the magnetic field [*Phys. Rev. B* **72**, 214432 (2005)]. Therefore, we assume that the field-dependent part of ΔR corresponds to ΔR_L . The measured ΔR_L shows

a threshold behavior that is a characteristic feature of the phase transition. We included this explanation in the lines 137~157 of the revised manuscript. We also included both the magnetic field independent and dependent parts of ΔR in Figure 3 of the revised manuscript.

Finally, the magnetic characterization of FeRh thin films in Suppl. Fig. 1b-1c shows an approximately 20% lower magnetization than expected for the fully ferromagnetic phase. Was an external magnetic field applied during the temperature loop? The authors might present a magnetic field induced hysteresis loop at 300 K to support their claim of negligible magnetization in the antiferromagnetic phase.

(Response) To set the direction of the magnetic moment of the ferromagnetic phase of FeRh, we applied an external field of 0.15 T, which is the maximum field in our optical setup.

We measured the hysteresis loop of the AFM phase of FeRh at 300 K and the FM phase of FeRh at 430 K using a vibrating sample magnetometer (Fig. R5). Whereas the FM phase shows a clear hysteresis with a saturation magnetization of 930 emu/cc ($0.93 \times 10^6 \text{ A m}^{-1}$), the AFM phase shows a negligible magnetization. We included this result in Supplementary Section 2.

Figure R5. Magnetization (M) versus magnetic field (B-field) of FeRh films. M of FeRh 20 nm film was measured using a vibrating sample magnetometer with the B-field along the in-plane direction. At a base temperature of 300 K, FeRh becomes complete AFM phase and shows no magnetization. At a base temperature of 430 K, FeRh becomes complete FM phase and shows saturation magnetization of 930 emu/cc ($0.93 \times 10^6 \text{ A m}^{-1}$) and coercivity field of $\sim 150 \text{ Oe}$.

In conclusion, although the observation of the angular momentum transfer from Cu upon the

phase transition in FeRh is intriguing, the discussion of this effect is not sufficiently deep, at times lacking context and references for the claims, and contains several inconsistencies which render the work inappropriate for publication in Nature Communications.

(Response) We understand that several different mechanisms, including Pressacco's paper, have been proposed for the underlying mechanism of the phase transition of FeRh. Although our work cannot solve this issue completely, we believe that our work provides important insight for the angular momentum transfer during the phase transition. We directly measured spin accumulation and show that polarization of the spin current is opposite for the ultrafast demagnetization and magnetization processes. Furthermore, we demonstrated the dM/dt model well explains the magnitude and timescale of the spin current. Despite these matchings between experiment and modeling, we acknowledge that our modeling cannot explain all the details during the phase transition, such as the sub-ps dynamics. To clarify the limitation of our modeling, we explicitly notified assumptions that we made in the lines 196~199 of the revised manuscript.

“For the spin generation, we make two assumptions: 1) the ps dynamics of the domain nucleation during the phase transition is mediated by the magnetic excitations, i.e., magnons; 2) a change of angular momentum of the magnon bath is quickly supplied by the conduction electron bath.”

We hope that our revised manuscript resolves major concerns of the reviewer.

Reviewers' Comments:

Reviewer #1:

Remarks to the Author:

The authors have fully addressed my criticism. I do confirm that the paper reports new and very intriguing data on a heavily debated topic - ultrafast kinetics of the phase transition in FeRh. Every single breakthrough in experimental investigation of the subject has been associated with the development of a new experimental technique. This is exactly the case in the present work which has combined spintronics and ultrafast pump-probe technique in one experiment. As the topic is heavily debated, practically all published papers on this subject are not "sufficiently deep" or overarching. The debate, which more and more resembles chicken-or-egg dilemma, urgently needs new input from experiments and this is exactly what this paper gives. I strongly recommend it for publication!

Reviewer #2:

Remarks to the Author:

I appreciate the authors' effort to address the reviewers' concerns and suggestions. Having considered the revised manuscript, I stand by the previous conclusion that the results are relevant for the understanding of ultrafast magnetic phase transitions, however, the discussion needs to be clarified before accepting the manuscript for publication. The remaining points in question are the following:

1. Regarding the link between the lattice expansion and magnetic field, the authors refer to the work of S. Maat et al., which among other findings demonstrates the offset of the phase transition temperature with applied magnetic field, under equilibrium conditions. Please note that the expected reduction of the equilibrium phase transition temperature for the applied field of 150 mT is 1.2 K (considering the reported offset of -8 K/T). Following this reasoning, it is not clear why the lattice expansion associated fast reflectivity change would be observable only under the condition of the applied magnetic field? Along with clarifying this point, it would help the understanding to specify how the raw signals with and without the applied field were processed, i.e., which of the R_A , R_T , R_L are present with and without the applied field. Significant changes in the optical reflectivity associated with the phase transition (see for instance the papers of V. Saidl et al. <https://doi.org/10.1088/1367-2630/18/8/083017> or S. Bennett et al. <https://doi.org/10.1364/OME.9.002870>, unfortunately not cited in the manuscript) do not necessarily depend on the magnetic field (the transition can be induced solely by temperature), which contradicts the repeated statement in Lines 146-147 of the revised manuscript.

2A. My point regarding the correlation of the MOKE and optical reflectivity signals at 2.5 ps is partially addressed. However, the assumption of the magnetic domains being partially aligned is quite bold, considering the nucleated phase domains are typically small enough to be below the superparamagnetic limit.

2B. Lines 156-157 "The same time delay of 2.5 ps in ΔM and ΔR_L suggests that the timescale of the domain nucleation during the phase transition is limited by the speed of the lattice expansion." Here, the domain nucleation coincides with the optical reflectivity signal. It is ambiguous whether the speed "limit" is structural or magnetic (i.e., emergence of the FM phase, regardless of the magnetization alignment).

2C. Related to the magnetization process itself, the phase transition proceeds mainly through phase domain nucleation and coalescence (see for instance Baldasseroni et al. <http://dx.doi.org/10.1063/1.4730957>), with limited DW motion. Line 111 and Fig. 5a should be adapted accordingly.

3. Acoustic wave propagation: What is the value of the d_{surf} parameter?

4. The measured M_s is about 20% lower than the value typically reported for FeRh thin films. What may be the cause of this discrepancy?

5. The manuscript uses SI units. However, the supplementary material uses cgs (Oe, emu/cc).

6. Labels on the y-axes of Fig. 2 and Fig. 4 seem to contain typographic errors (numbers mixed in).

Response letter to Referee's comments

Manuscript NCOMMS-22-36165A

"Spin current driven by ultrafast magnetization of FeRh" by Kyuhwe Kang *et al.*

We thank both referees for their constructive comments. Below, we summarize the provide point-by-point responses to the referee's comments and suggestions that requires responses. The corresponding corrections are incorporated in the revised manuscript with track changes. In addition, we included Daniel Yesudas as a co-author of this work. He helped us to find a growth condition of FeRh. Although the actual sample growth was led by Hiroki Omura and Kyuhwe Kang, Daniel Yesudas also did some contribution.

Reviewer #1 (Remarks to the Author):

The authors have fully addressed my criticism. I do confirm that the paper reports new and very intriguing data on a heavily debated topic - ultrafast kinetics of the phase transition in FeRh. Every single breakthrough in experimental investigation of the subject has been associated with the development of a new experimental technique. This is exactly the case in the present work which has combined spintronics and ultrafast pump-probe technique in one experiment. As the topic is heavily debated, practically all published papers on this subject are not "sufficiently deep" or overarching. The debate, which more and more resembles chicken-or-egg dilemma, urgently needs new input from experiments and this is exactly what this paper gives. I strongly recommend it for publication!

Reply) We thank the reviewer for the high evaluation of our work.

Reviewer #2 (Remarks to the Author):

I appreciate the authors' effort to address the reviewers' concerns and suggestions. Having considered the revised manuscript, I stand by the previous conclusion that the results are relevant for the understanding of ultrafast magnetic phase transitions, however, the discussion

needs to be clarified before accepting the manuscript for publication. The remaining points in question are the following:

1. Regarding the link between the lattice expansion and magnetic field, the authors refer to the work of S. Maat et al., which among other findings demonstrates the offset of the phase transition temperature with applied magnetic field, under equilibrium conditions. Please note that the expected reduction of the equilibrium phase transition temperature for the applied field of 150 mT is 1.2 K (considering the reported offset of -8 K/T). Following this reasoning, it is not clear why the lattice expansion associated fast reflectivity change would be observable only under the condition of the applied magnetic field? Along with clarifying this point, it would help the understanding to specify how the raw signals with and without the applied field were processed, i.e., which of the R_A , R_T , R_L are present with and without the applied field. Significant changes in the optical reflectivity associated with the phase transition (see for instance the papers of V. Saidl et al. <https://doi.org/10.1088/1367-2630/18/8/083017> or S. Bennett et al. <https://doi.org/10.1364/OME.9.002870>, unfortunately not cited in the manuscript) do not necessarily depend on the magnetic field (the transition can be induced solely by temperature), which contradicts the repeated statement in Lines 146-147 of the revised manuscript.

Reply) Indeed, the magnetic-field-dependent part of ΔR is about ten times smaller than the magnetic-field-independent part of ΔR . So, the raw ΔR is dominated by the magnetic-field-independent part. The raw ΔR already has the ΔR_L contribution in the nonlinear dependence on the pump fluence. (ΔR_T and ΔR_A should depend on the pump fluence linearly.) To extract the magnetic-field-dependent part of ΔR , we took the difference of ΔR without and with magnetic field of 0.15 T. Note that this magnetic-field-dependent part corresponds to a partial change of ΔR_L by 0.15 T, but it effectively excludes ΔR_T and ΔR_A . We included this explanation on the line 148~161 of the revised manuscript. To emphasize that the magnetic-field-dependent part is much smaller than the raw signal, we show the raw ΔR in Fig. 3a.

2A. My point regarding the correlation of the MOKE and optical reflectivity signals at 2.5 ps is partially addressed. However, the assumption of the magnetic domains being partially aligned is quite bold, considering the nucleated phase domains are typically small enough to be below the superparamagnetic limit.

Reply) The assumption for the magnetic alignment of the nucleated domains is critical for our analysis of the spin generation and transport (Fig. 5). When the nucleated domains have random orientations in magnetization, the magnon bath has no net change of angular momentum. We want to state this assumption explicitly so that readers can judge whether this assumption is valid or not.

2B. Lines 156-157 "The same time delay of 2.5 ps in ΔM and ΔR_L suggests that the timescale of the domain nucleation during the phase transition is limited by the speed of the lattice expansion." Here, the domain nucleation coincides with the optical reflectivity signal. It is ambiguous whether the speed "limit" is structural or magnetic (i.e., emergence of the FM phase, regardless of the magnetization alignment).

Reply) The causal relation between ΔM and ΔL is a chicken-and-egg problem for the mechanism of the phase transition in FeRh. We argue that the speed limit of the phase transition originates from the structural dynamics because the 2.5 ps is half of the acoustic echo at 5 ps. However, if one can find an explanation for 2.5 ps in terms of the magnetic dynamics, such as speeds of precession or domain wall motion, the speed limit could be related to the magnetic origin. We included this explanation on the line 162~166 of the revised manuscript.

2C. Related to the magnetization process itself, the phase transition proceeds mainly through phase domain nucleation and coalescence (see for instance Baldasseroni et al. <http://dx.doi.org/10.1063/1.4730957>), with limited DW motion. Line 111 and Fig. 5a should be adapted accordingly.

Reply) We modified the line 111 and Fig. 5a to note that the sub-ns dynamic is related to the domain growth as well as coalescence process. We also cited Baldasseroni's paper as Ref. [37].

3. Acoustic wave propagation: What is the value of the d_{surf} parameter?

Reply) We expect that d_{surf} is close to a light penetration depth of ~ 10 nm. Assuming a fixed v_s of 5 km/s, d_{surf} of 7 nm can explain the acoustic echo at 5 ps.

4. The measured M_s is about 20% lower than the value typically reported for FeRh thin films.

What may be the cause of this discrepancy?

Reply) As far as we know, the highest M_s of FeRh among the previous reports is around 1100 emu/cc. We do not clearly understand why our sample show a somewhat smaller M_s of 930 emu/cc. We expect that M_s depends on the ordering degree, concentration of Rh, thickness, strain, and so on.

5. The manuscript uses SI units. However, the supplementary material uses cgs (Oe, emu/cc).

Reply) We change the cgs unit in the supplementary materials to the SI unit.

6. Labels on the y-axes of Fig. 2 and Fig. 4 seem to contain typographic errors (numbers mixed in).

Reply) There are no typographic errors in Fig. 2 and Fig. 4. Some panels of these figures have double y-axes. For example, panels of a, b, and c of Fig. 2 have the left y-axis in the unit of the Kerr rotation and the right y-axis in the unit of the relative magnetization.

Reviewers' Comments:

Reviewer #2:

Remarks to the Author:

The authors have addressed the major ambiguous points raised in the previous review round. I appreciate their effort to improve the discussion of these findings critical for understanding of ultrafast metamagnetic phase transitions. Hence, I support publication of the manuscript in Nature Communications.

Regarding point 6, the incorrect labels were likely caused by compilation to pdf, as these do not occur in the manuscript .docx file.

Response to reviewers

REVIEWERS' COMMENTS

Reviewer #2 (Remarks to the Author):

The authors have addressed the major ambiguous points raised in the previous review round. I appreciate their effort to improve the discussion of these findings critical for understanding of ultrafast metamagnetic phase transitions. Hence, I support publication of the manuscript in Nature Communications.

Regarding point 6, the incorrect labels were likely caused by compilation to pdf, as these do not occur in the manuscript .docx file.

Response to reviewer

In the previous review, the reviewer#2 mentioned that labels on the y-axes of figures 2 and 4 have a typographic error. However, as the reviewer pointed out in this review, there is no error in the word docx file. After confirming no errors, we submit the figure files in a pdf format.